# Polish medical students facing the pandemic—Assessment of resilience, well-being and burnout in the COVID-19 era

Joanna Forycka[☯], Ewa Pawłowicz-Szlarska[iD][☯], Anna Burczyńska, Natalia Cegielska, Karolina Harendarz, Michał Nowicki*

Department of Nephrology, Hypertension and Kidney Transplantation, Medical University of Lodz, Lodz, Poland

☯ These authors contributed equally to this work.
* nefro@wp.pl

## Abstract

### Introduction

Recent reports indicate that COVID-19 pandemic has significant influence on medical professionals' mental health. Strict limitations in clinical practice and social interactions within academic community, which had to be introduced, could lead to significant psychological distress in medical students. The aim of the study was to assess resilience, well-being and burnout among Polish medical students in the COVID-19 era.

### Methods

The online survey consisting of validated questionnaires assessing resilience (Resilience Scale 14; RS-14), well-being (Medical Student Well-Being Index) and burnout (Maslach Burnout Inventory) as well as self-created survey concerning mental health problems, use of stimulants, SARS-CoV-2 infection, work in COVID-19 units, medical education and social attitude towards health care professionals in the pandemic era was distributed via Facebook and other online students' platforms. 1858 MSs from all polish medical schools agreed to fill in the survey.

### Results

'Very low', 'low' and 'on the low end' levels of resilience were found in 26%, 19.1% and 26.9% of the study group, respectively. Students with higher resilience level presented better attitude towards online and hybrid classes. 16.8% of respondents stated that they worked, currently work or plan to work voluntarily at the pandemic frontline. In terms of burnout, these respondents presented lower exhaustion (p = 0.003) and cynicism (p = 0.02), and higher academic efficacy (p = 0.002). That group also showed greater resilience (p = 0.046). The SARS-CoV-2 infection among respondents, their relatives and friends did not influence the results. 39.1% of respondents declared the need of the psychological or psychiatric consultation in relation to pandemic challenges. 231 (26.4%) participants previously diagnosed with mental health disorders noticed worsening of their symptoms. Increased intake of

**Data Availability Statement:** We share the dataset as the supporting file. Due to license agreements of the psychometric instruments used in the study we

have blinded the headings of some columns since we cannot share publicly questions of these surveys. The full access to psychometric instruments protected by license agreements used in the study (Maslach Burnout Inventory – General Survey for Students, The Resilience Scale 14 and Medical Students Well-Being Index) may be gained via websites https://www.mindgarden.com/, https://www.resiliencecenter.com/products/resilience-scales-and-tools-for-research/the-rs14/ and https://www.mywellbeingindex.org/versions/medical-student-well-being-index, respectively. The license agreement allows us to use these tools, but we cannot post the exact questions, but they are available to anyone interested on the websites of the respective owners.

**Funding:** This study was supported by Medical University of Lodz (503/1-151-02/503-01). No external funding was received.

**Competing interests:** The authors have declared that no competing interests exist.

alcohol, cigarettes or other stimulants was noticed by 340 (28.6%) respondents. 80.2% of respondents thought that social aversion and mistrust towards doctors increased during the pandemic and part of them claimed it affected their enthusiasm toward medical career.

## Conclusions

The majority of medical students presented low levels of resilience and high burnout at the time of pandemic. Providing necessary support especially in terms of mental health and building up the resilience of this vulnerable group seems crucial to minimize harm of current pandemic and similar future challenges.

## Introduction

The overwhelming scale of the global crisis and the number of problems faced by all countries with the SARS-CoV-2 outbreak such as hospital overload, economic crisis, income instability, social isolation, constant anxiety about rapidly changing circumstances, and an unpredictable future are the cause of exhaustion in society as a whole. Medical students, who are more vulnerable to experiencing psychological distress than the general and peer population, had to combat an additional specific set of challenges connected with their studies and professional future—e.g., helping in medical facilities [1, 2] which posed a risk of infection and transmitting the virus to loved ones. A major challenge was also halting clinical training [2] which is crucial for an effective medical education, and moving to e-learning as the leading form of medical education during the pandemic [3, 4]. Besides, the closing of universities and public libraries may have forced some students into an inadequate learning environment [5].

The pandemic extended the role of online learning in medical education. It was reported that prolonged exposure to a computer or other device was associated with burnout, increased levels of stress, and the occurrence of stress-related mental and physical symptoms, which has an impact on an individual's quality of life and daily activities. Furthermore, it can be related to the development of mental health problems in students including moderate-to-severe depression [6]. Universities have been forced to strictly limit students' clinical practice and work placements due to the pandemic. Extra-curricular activities like attending conferences and conducting one's own research were also constricted which may have resulted in worries about career development [7]. These new difficult circumstances tested medical students' perseverance, stress management, and coping abilities. However, besides all the inadequacies associated with online education, it must be indicated that in these exceptional circumstances, it allowed communication with relatives and friends and contact with the academic community, enabling learning and teaching.

The first case of COVID-19 in Poland was confirmed on the 4th of March 2020. Restrictions were implemented by virtue of the statute signed by the President of Poland on the 7th of March 2020. All mass events were canceled. Schools and universities were closed, and online learning was recommended. Crossing borders was restricted, with obligatory quarantine for every in-coming person. In the following days, parties and private meetings were forbidden. It was strongly recommended by the government to stay at home and leave only when essential. There was also a limited number of people allowed to stay in closed spaces like shops or means of public transportation. Walking in parks, woods, or on beaches or boulevards was also forbidden. Hairdressers, beauty salons, rehabilitation centers, and all shops except for grocery

stores, pharmacies, and newsagents were closed. In April 2020, the cabinet council instituted regulations that obligated everyone to cover their mouth and nose in public spaces.

It may be hypothesized, that these extraordinary actions such as social distancing and closing schools, offices, and entertainment venues might have taken a toll especially on those more susceptible to stress and less resilient individuals.

Resilience is defined as the process of, capacity for, or outcome of successful adaptation despite challenging or threatening circumstances [8], hence an ability for flexible adaptation to challenges [9]. It has an influence on other terms pertaining to an individual's or a group's mental state—well-being and burnout [10, 11]. Well-being is successful and fulfilling performance in the psychological and physical domains of life [12, 13]. It is a state in which a person is aware of their own abilities, can cope with stress, and can work efficiently. The World Health Organization considers well-being to be an integral part of health [14]. Burnout is a syndrome of emotional exhaustion, cynicism, and poor satisfaction caused by occupation-related stressors [15]. Our study assesses resilience, well-being, and burnout among medical students in the COVID-19 era due to the new challenges brought by the pandemic. Several compounding factors like mental health problems, use of stimulants, medical education in the COVID-19 era, the level of mistrust towards healthcare professionals, the presence of SARS-CoV-2 infections in respondents, their families, and friends, and volunteer or paid work in COVID-19 units were taken into account.

## Methods

### Study survey

A web-based survey hosted on the Survey Monkey application was distributed via various student Facebook groups and Instagram profiles (known as *studygrams*) assembling Polish medical students. The survey was sent to student council presidents of Polish medical schools to share with all the medical students from their universities, and to Polish students acting as influencers and promoting health and medical knowledge through their Instagram and Facebook profiles. All Facebook groups were open for medical students from the particular university, and Instagram profiles—for subscribers to the profile. The survey was conducted from the 5th January 2021 to the 6th February 2021 and posted once on all of the above-mentioned groups. The approximate time to complete the survey was expected to be 10–15 minutes. Due to the method of data collection (no computer IP numbers were collected) we could not prevent the same person filling out the questionnaire more than once.

The first page of the survey contained the informed consent. Expressing consent enabled proceeding to the next questions in the survey.

The study survey consisted of three validated questionnaires—Resilience Scale-14, Medical Students Well-Being Index, and Maslach Burnout Inventory—General Survey for Students, as well as self-created questions. All validated questionnaires were used in accordance with the license agreements.

The Resilience Scale (RS-14) is a 14-item survey validated in the population of Polish young adults (age 19–27) [16]. RS-14 uses a Likert scale with seven possible responses for each item ranging from 1 (strongly disagree) to 7 (strongly agree). All items are positively worded, and a higher score indicated greater resilience. RS-14 scores range from 14 to 98. The RS-14 score ranges indicate the level of resilience, as follows: very low resilience (14–56 points), low (57–64), on the low end (65–73), moderate (74–81), moderately high (82–90) and high (91–98 points). The RS-14 scores correlate significantly with measures of positive concepts (i.e., life satisfaction). Resilience was negatively related to indexes of perceived stress and the dimension of depression [8].

Medical Students Well-Being Index (MSWBI) is a validated tool in a medical student population and consists of 7 "yes or no" questions evaluating distress across a variety of dimensions including fatigue, depression, burnout, anxiety, stress, and mental as well as physical quality of life [17]. MSWBI scores range from 0 to 7, and 7 points indicate the greatest level of distress. The MSWBI was proven to be a useful tool in identifying students with severe distress, and the MSWBI scores correlate with quality of life, fatigue, recent suicidal ideation, burnout, and the likelihood of seriously considering dropping out of medical school [13].

The 16-item Maslach Burnout Inventory—General Survey for Students (MBI-GS(S)) is a validated tool used to measure burnout among student populations. The MBI-GS(S) (Mind Garden Inc., Menlo Park, CA), designed for college and university students, is divided into three distinct domains: emotional exhaustion (EE)—5 items, cynicism (CY)—5 items, and academic efficacy (AE)—6 items [15, 18]. The respondents were asked to report the frequency of particular feelings associated with their studying, and each statement was assessed on a time scale (never—0 points, a few times a year—1 point, once a month or less—2 points, a few times a month—3 points, once a week—4 points, a few times a week—5 points, and every day —6 points). By adding points for items in particular dimensions, burnout was classified as low, moderate, and high in all three dimensions (see Table 4 for reference values). High scores for CY and EE and low scores for AE meant greater burnout (three-dimensional burnout). Two-dimensional burnout has been described previously in the literature [19] and is defined as high cynicism and high emotional exhaustion. Since there was no Polish version of the MBI-GS(S) available, the authors of this study requested permission from Mind Garden Inc. to create a Polish translation. The translation was prepared, evaluated, and is now made available to all researchers via the Mind Garden platform (https://www.mindgarden.com/).

The questions in the self-created part of the survey aimed to assess:

1. The psychological condition of the respondent—diagnosis of psychological or psychiatric disorders, psychological care or psychiatric treatment, use of psychiatric drugs, OTC drugs (especially tranquilizers), use of alcohol, cigarettes, and other stimulants, change in self-esteem during the pandemic;

2. SARS-CoV-2 infections in respondents and/or their families as well as the clinical course of the infection;

3. COVID-19-related volunteer or paid work—extracurricular volunteer or paid work in the COVID-19 units as hospital orderlies, attitude and willingness to work after classes, or to volunteer during class hours, with COVID-19 patients instead of participating in online education, the level of fear connected with working or volunteering and the reasons behind it;

4. Medical education during the pandemic—the number of in-person classes, including practical and clinical classes, participation in summer clinical clerkships, evaluation of the online education—the preferred and most frequent form of online classes and the personal assessment of the impact of online education on practical skills, theoretical knowledge, and future work; the level of motivation, procrastination, and solitude during the pandemic;

5. Student views on cases of social aversion and mistrust towards healthcare professionals during the pandemic and their importance for the motivation to continue medical education.

The self-created part of the survey is provided as the (S1 File).

The pilot study of the self-created part of the survey was performed on a group of 25 students of different years, and their remarks on the quality of the questions were applied in the final version of the survey.

The demographic data collected as part of the survey included age, gender, year of study, and name of medical school.

All questions, except for informed consent, were answered voluntarily.

The study protocol was approved by the local ethics committee of the Medical University of Lodz.

## Study group

Medical students of all years (1st–6th) from all 22 Polish medical schools (medical universities or medical faculties) that have the same formats of curriculum consisting of pre-clinical classes (anatomy, physiology, etc.), clinical surgical and non-surgical subjects, and summer clinical clerkships were eligible to complete the survey. Inclusion criteria were as follows: studying medicine at one of the Polish medical schools and consent to participate in the study. Out of 1,864 respondents who entered the survey, 1,858 respondents gave their consent. 55.5% of the respondents answered all questions. The average time to complete the whole survey was 7 minutes and 53 seconds. The study group characteristics are provided in Table 1.

## Statistical analysis

Results are presented as mean ± standard deviation (SD) or median and interquartile range (IQR) depending on the normality of the distribution of each variable. Percent values are given in relation to the number of respondents who answered the particular question.

Statistical analysis was performed using *Statistica* ver. 13.1 PL software. Graphs were plotted with MS Excel and *Statistica*. T-test was used for comparisons between two independent groups. ANOVA and post-hoc tests were applied for comparisons of more than two groups. The Chi-square test was used for comparisons of categorical data. Correlations were assessed with Pearson's method. Pairwise deletion of missing data was applied.

**Table 1. The study group characteristics (N = 1,858).**

| Characteristic (number of responses) | Number of respondents (%) |
|---|---|
| **Gender (N = 1,847)** | |
| Male | 407 (22%) |
| Female | 1,435 (77.7%) |
| Other | 5 (0.3%) |
| **Age, years (N = 1,852)** | |
| 18–20 | 473 (25.5%) |
| 21–23 | 949 (51.3%) |
| 24–26 | 378 (20.4%) |
| >26 | 52 (2.8%) |
| **Year of study (N = 1,850)** | |
| 1st | 325 (17.6%) |
| 2nd | 384 (20.8%) |
| 3rd | 411 (22.2%) |
| 4th | 313 (16.9%) |
| 5th | 248 (13.4%) |
| 6th | 169 (9.1%) |

## Results

### Mental health condition

478 (40.2%) respondents stated that they had sought the help of a psychologist or psychiatrist in the past, and 463 (39.1%) confirmed that they noticed a need for a psychological or psychiatric consultation in relation to pandemic challenges (social isolation, restrictions, and fear of SARS-CoV-2 infection). There were 203 students who did not seek psychological help in the past but did feel such a need at the time of the pandemic. 280 (23.5%) respondents took medications prescribed by a psychiatrist, 558 (47%) admitted that they took over-the-counter (OTC) anti-anxiety medications during studies, and 323 (27.9%) reported an increase in the doses of these medications during the pandemic. As for diagnosed psychiatric conditions, 133 (11.2%) respondents were diagnosed with depressive disorders, 76 (6.4%)—anxiety disorders, 44 (3.7%)—stress-related disorders, 16 (1.3%)—personality disorders, and 2 (0.2%)—psychotic disorders; 28 respondents choose the option "other disorders". 231 (26.4%) participants previously diagnosed with such disorders noticed worsening of their symptoms. 665 (56%) participants stated that the pandemic impacted their self-esteem negatively, 394 (33.2%) noticed no impact, and 128 (10.7%) declared a positive impact. 340 (28.6%) respondents declared that during the pandemic, they used alcohol, cigarettes, or other stimulants more often than before.

### SARS-CoV-2 infection

237 (19.9%) respondents were infected with SARS-CoV-2 virus. Out of this group, in 39 (16.5%) the infection was asymptomatic, in 179 (75.5%) the course of the disease was mild, and in 19 (8%)—severe. 901 (75.8%) participants stated that at least one member of their family or a friend was infected with the coronavirus, and 132 (11.1%) confirmed that at least one of these people died in the course of COVID-19. The SARS-CoV-2 infection, both among respondents, their friends, and relatives did not influence the results of the survey.

In general, in Poland, there were 2,9 million confirmed cases and 75,600deaths due to COVID-19 by the end of September 2021.

### Volunteer or paid work in COVID-19 units

93 participants worked, 31 currently work and 139 plan to start work voluntarily in the COVID-19 healthcare units. Asked about their attitude towards referral to work in the COVID-19 healthcare units, 505 (43.4%) stated that they would only do so to avoid potential consequences such as a financial penalty in the case of resignation from a previously accepted job or fear of condemnation by university authorities, 342 (29.4%) answered that they would be happy to help, 301 (25.9%) treat such work as their duty, and only 16 (1.3%) would even consider dropping out of medical school to avoid such work. 772 participants stated that they have several concerns regarding such voluntary work, these concerns are presented in Table 2. Only 125 (10.5%) participants stated that they are practically and theoretically well-prepared

**Table 2. Concerns regarding voluntary work in the COVID-19 healthcare units perceived by the study participants.**

| Concerns regarding voluntary work in the COVID-19 healthcare units | Number of respondents (%) |
| --- | --- |
| possibility of infection and transmission of infection to loved ones | 610 (79%) |
| spending time volunteering instead of studying | 533 (69%) |
| fear of liability and potential consequences for providing help incorrectly | 483 (62.6%) |
| insufficient supply of the personal protective equipment | 429 (55.6%) |

to take such a job, and this was significantly associated with the year of studies (Mann Whitney U test p<0.001).

## Medical education in the pandemic era

The most commonly applied method of e-learning was live classes via Internet communicators, as reported by 956 participants (80.6%), other e-learning methods comprised recorded lectures and/or seminars, presentations shared with students, and exercises for their own work. The two main methods of e-learning that were preferred most by respondents were recorded lectures and/or seminars (preferred by 40.3% of students) and live classes via Internet communicators (preferred by 39.2% of students). Practical (i.e., laboratory classes, simulations, practical classes in anatomy) and clinical classes (classes in in-patient or out-patient settings) in the first semester of the academic year 2020/2021 were completed on a limited basis.

The vast majority of students expressed concerns over their practical skills (84.3%) and the level of theoretical knowledge (66.6%) after some months of mostly online teaching. 59.9% are worried about their performance at the Final Medical Exam—passing this exam is compulsory in order to receive a medical license, also the result of the exam is the application criterion for residency.

Students' attitudes to online learning, motivation to learn, and behavior patterns related to novel situations in medical education were also assessed. 806 (68%) respondents confirmed that they delay their tasks and postpone their duties more often in the online teaching era. Reduced motivation to learn was reported by 934 (78.8%) participants. 596 respondents stated that they learn less than before the pandemic, 338—comparably, and 250 learn more. The level of solitude of students was assessed on a 5-point Likert scale with 1 point reflecting no solitude at all and 5 points reflecting overwhelming solitude; the median was 4 (IQR 3).

## Social attitude towards healthcare professionals during the pandemic

949 (80.2%) of respondents thought that social aversion and mistrust towards doctors increased during the pandemic, and 43.3% of this group confirmed that this may affect their enthusiasm toward a future medical career. Also, the challenges faced by the national healthcare system (such as a shortage of healthcare professionals, shortage of medical equipment, and the unwillingness of society to undertake precautionary measures against COVID-19) negatively affect attitudes toward working as a doctor, as reported by 43.8% of participants. The above-mentioned issues were related to considerations and plans to practice medicine abroad, confirmed by 39.1% of respondents.

## Resilience

A total of 1032 respondents answered all RS-14 questions. The mean resilience score in this group was 65.5±13.6, the lowest score was 25, and the highest—98 (RS-14 scores from 14 to 98). The percentage of respondents presenting a specific level of resilience is shown in Fig 1.

Worthy of note is that 72% of the study population presented lower levels of resilience ('very low', 'low', 'on the lower end'), indicating a decreased ability to adapt to challenging circumstances.

RS-14 scores depending on medical student characteristics gathered in the self-created survey are provided in Table 3.

Resilience was not dependent on age; however, the highest resilience level was found among 4th-year students and the lowest among 1st- and 6th-year students (Fig 2).

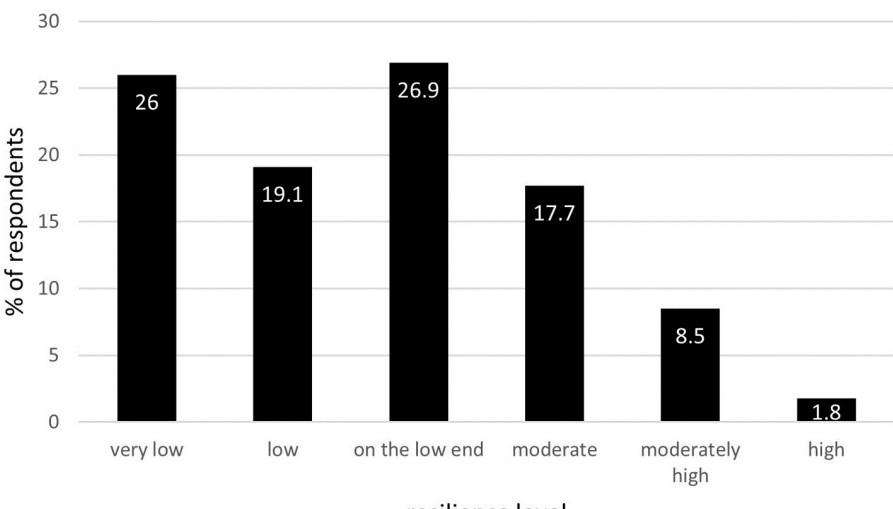

**Fig 1. The percentage of respondents presenting particular levels of resilience according to Resilience Scale-14 (N = 1,032).**

## Burnout

A total number of 1311 students answered all MBI-GS(S) questions. Mean scores, which may range from 0 to 6 for each dimension, were 3.9±1.4, 3.5±1.6, and 2.9±1.1 for EE, CY, and AE, respectively. The prevalence of high, moderate, and low levels of burnout in the three

**Table 3. RS-14 scores depending on medical student characteristics gathered in the self-created survey.**

| | RS-14 scores | | | p-value |
|---|---|---|---|---|
| gender | men | | women | <0.01 |
| | 67.5±13.9 | | 64.6±13.2 | |
| diagnosed mental health conditions | yes | | no | <0.001 |
| | 59.7±13.8 | | 67.3±12.9 | |
| more often use of alcohol, cigarettes or other stimulants during the pandemic | yes | | no | |
| | 62±13.4 | | 66.8±13.4 | <0.001 |
| willingness to volunteer in the COVID-19 health-care units | yes | | no | 0.05 |
| | 67.8±12.5 | | 65±13.7 | |
| concerns on the impact of the online learning on the level of knowledge | yes | | no | <0.01 |
| | 64.5±13.7 | | 67.6±12.9 | |
| concerns on the impact of the online learning on the result of Final Medical Exam | yes | | no | |
| | 64.2±13.9 | | 67.3±13.1 | <0.01 |
| reduced motivation in the online learning era | yes | | no | <0.01 |
| | 64.6±13.4 | | 68.7±13.5 | |
| increased motivation in the online learning era | yes | | no | <0.001 |
| | 63.7±13.1 | | 69.3±13.7 | |
| susceptibility to the cases of social mistrust | yes | | no | 0.02 |
| | 64.9±13.6 | | 67.9±13.1 | |
| low enthusiasm towards future medical career | yes | | no | 0.07 |
| | 63.8±13.2 | | 66.8±13.9 | |
| impact of pandemic on the self-esteem | negative | neutral | positive | <0.001 |
| | 61.5±13.2 | 69.4±12.4 | 71.6±12.6 | |

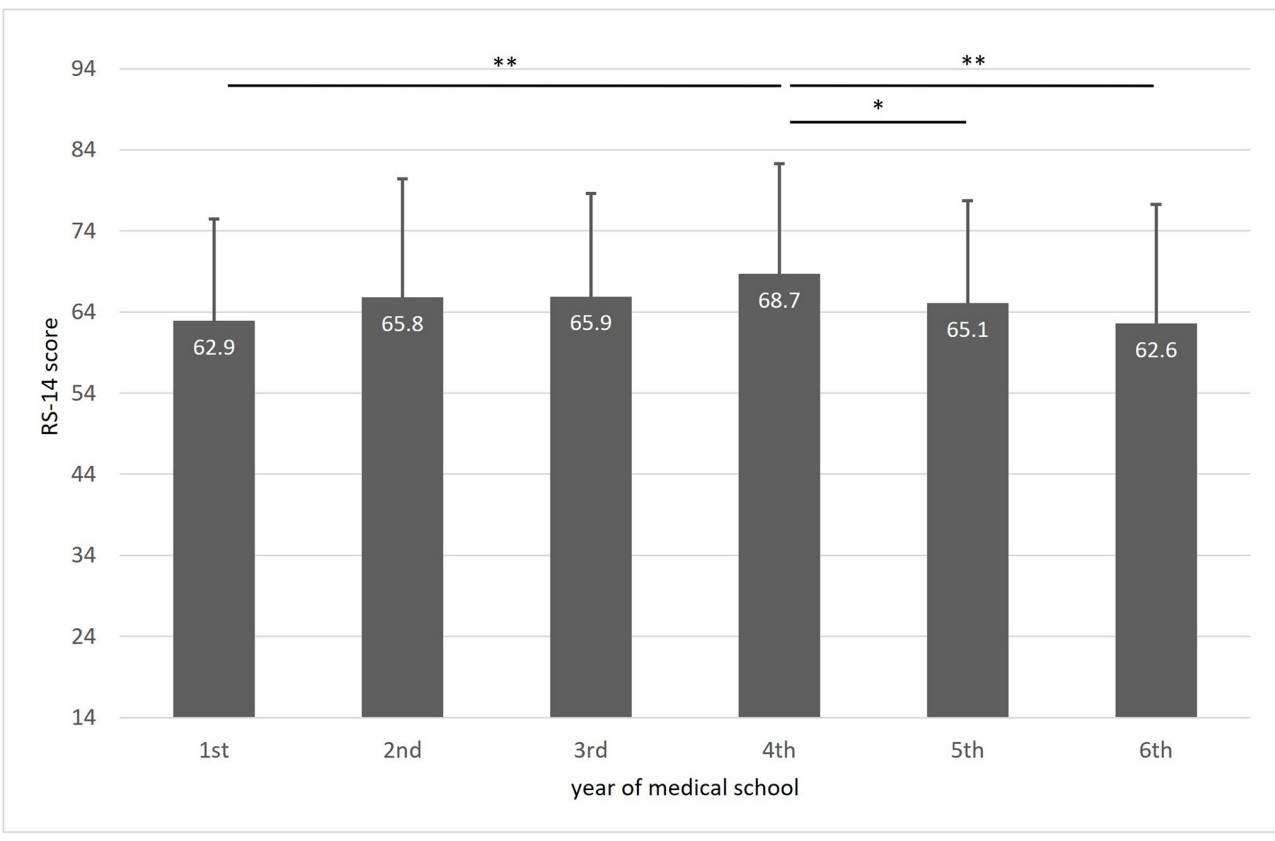

**Fig 2. Mean RS-14 scores among students of different years of medical school.** * p<0.05; ** p<0.01.

dimensions is provided in Table 4. Average scores for the three dimensions of burnout among students of different years of medical school are provided in Fig 3.

According to the two-dimensional (high EE + high CY) and three-dimensional (high EE + high CY + low AE) criteria, the presence of the overall burnout was determined (Table 4).

Participants who reported diagnosed mental conditions presented more severe burnout in all three dimensions (4.2±1.9 vs. 3.8±1.1; p<0.001 for EE; 3.7±1.6 vs. 3.4±1.1; p<0.001 for CY; 2.8±1.1 vs. 2.9±1; p = 0.02 for AE). Burned-out students (three-dimensional criterion) used stimulants more often than those who did not present burnout (Chi$^2$ Pearson p = 0.01). The willingness to volunteer was declared by students with significantly lower burnout in all dimensions.

The students diagnosed with burnout reported reduced motivation to learn significantly more often (Chi-square Pearson p<0.001) and increased procrastination (Chi-square Pearson p<0.001) in the online learning arena. There is also a significant negative correlation between burnout and resilience (r = -0.38, p<0.05). Mean RS-14 scores for participants presenting burnout in particular dimensions at low, moderate, and high levels as well as for those presenting three-dimensional burnout are provided in Fig 4.

## Well-being

A total number of 1359 students answered all MSWBI questions And the median MSWBI value was 5 points. 79 (5.8%) respondents scored 7 MSWBI points, while the median MSWBI score for students of years 1–3 and 6 was 5 points, and for students of years 4 and 5–4 points.

**Table 4. Prevalence of high, moderate and low levels of burnout in particular dimensions and of the overall burnout according to the two- and three-dimensional criteria (N = 1,311).**

| Burnout level in particular dimension (reference values) | Number of respondents (%) |
|---|---|
| **Emotional exhaustion** | |
| low (0–7) | 78 (6%) |
| moderate (8–15) | 281 (21.4%) |
| high (≥16) | 952 (72.6%) |
| **Cynicism** | |
| low (0–5) | 126 (9.6%) |
| moderate (6–12) | 261 (19.9%) |
| high (≥13) | 924 (70.5%) |
| **Academic efficacy** | |
| low (0–23) | 1,084 (82.7%) |
| moderate (24–29) | 183 (13.9%) |
| high (≥30) | 44 (3.4%) |
| **Two—dimensional burnout**[a] | **Number of respondents (%)** |
| Yes | 786 (59.9%) |
| No | 525 (40.1%) |
| **Three—dimensional burnout**[b] | **Number of respondents (%)** |
| Yes | 711 (54.2%) |
| No | (45.8%) |

[a]. High emotional exhaustion + high cynicism.

[b]. High emotional exhaustion + high cynicism + low academic efficacy.

Correlations between MSWBI and burnout and resilience scores are provided in Table 5. Median MSWBI was 5 points for the group with three-dimensional burnout and 4 points for the group in which three-dimensional burnout was not found.

## Discussion

Our study addressed the ability of medical students to cope with challenges during the COVID-19 pandemic via assessing their resilience level. Additionally, we took burnout and well-being into account as reliable measures of general and education-related welfare. While medical students were already experiencing high levels of distress before the pandemic [20], these new challenges showed the priority of providing support and building resilience in this vulnerable population, with the aim being to have a workforce that will be able to combat the long-lasting consequences of the COVID-19 pandemic and who can effectively face similar crises in the future. The strict limitations on students' clinical practice and work placements imposed due to the pandemic caused decreased motivation and concerns of inappropriate levels of knowledge as well as questioning their abilities to pass final examinations and work as doctors [21], something that was also reported in our study. Almost two-thirds of the respondents presented low levels of resilience, indicating a problem with their ability to maintain or regain mental health while experiencing adversity [22] such as a global pandemic. Low resilience was correlated with more severe burnout, poor well-being, reduced motivation, and higher usage of stimulants during the COVID-19 pandemic. This may support the conclusion that building resilience is one of the components of creating driven, invested, and content healthcare professionals. Interestingly, the lowest resilience was found among the 1st- and 6th-year medical students, which may indicate that programs for building resilience should be

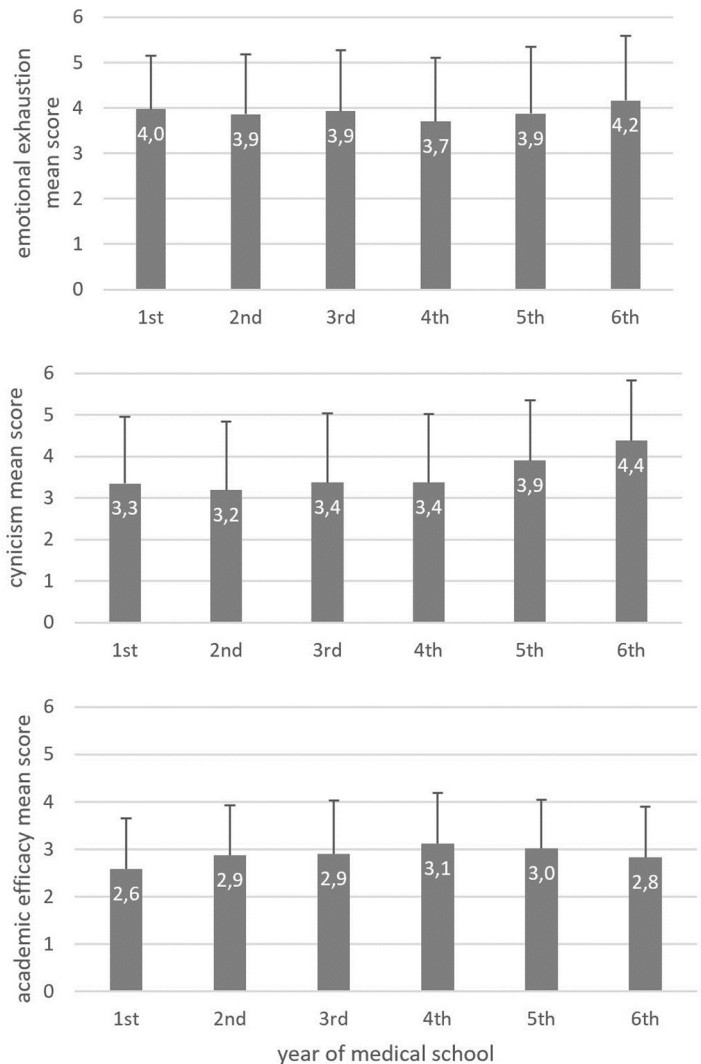

**Fig 3. Mean burnout scores in each dimension among students of different years of medical school.**

adjusted and changed over time. The need to prioritize healthy coping methods was reported in several other studies [23–25], and previous research has shown that medical students were eager to learn about healthy ways of coping with stress and to find out about tools that may help them with building resilience [26]. Jensen et al. identified many ways of building resilience in physicians—maintaining an interest in their role, acceptance of personal limitations, setting limits, or group practice [27]. Zwack et al. reported that leisure-time activity to reduce stress, cultivation of relations with family and friends, self-organization, and spiritual practices were among the methods that helped physicians to maintain high resilience [28]. These coping strategies could also be adapted to the population of medical students—courses in stress management and maintaining mental health, held by both professional psychologists as well as more experienced colleagues, could have a positive influence on building their resilience. Studies have pointed to self-efficacy, described as confidence in one's [own] abilities to successfully perform a particular behavior, influence events, and affect other lives, as a possible focus for building resilience in students [29–31].

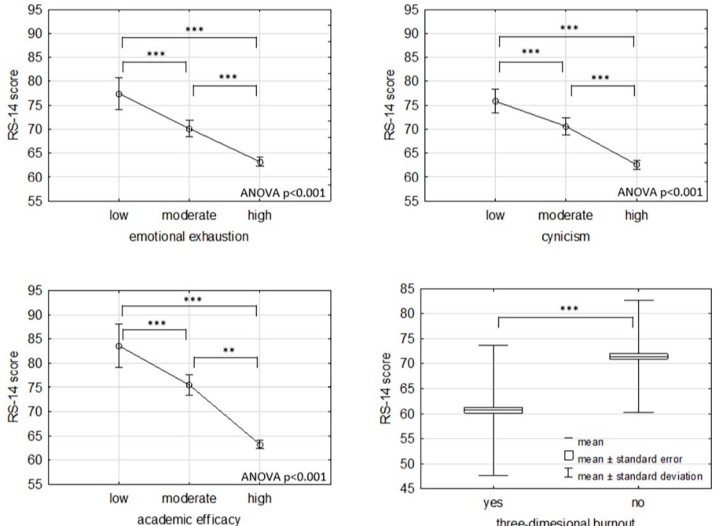

**Fig 4. Mean RS-14 scores for participants presenting burnout in particular dimensions at low, moderate and high levels (ANOVA) and mean RS-14 scores for participant demonstrated three-dimensional burnout'.** ** $p < 0.01$; *** $p < 0.001$.

Due to the combination of COVID-19 restrictions and online learning, researchers were forced to distribute questionnaires online, without direct contact with responders, which may have resulted in bias typical for online surveys. Sampling bias, defined as bias in which a sample is collected in such a way that some members of the intended population have a lower or higher sampling probability than others, must, therefore, be considered in our population—students who were more active on the social media had higher sampling probability, and some characteristics of this group may influence the results. Besides, desirability bias, which reflects the human tendency to appear and behave desirably and avoid undesirable traits, could influence our data, especially with regard to questions on volunteer work or learning habits. Nevertheless, this kind of data collection enabled reaching out to many medical students from different universities; however, survey questions other than validated questionnaires were created by the authors who, being members of the academic community, might allow their own experiences to lead to author bias. Additionally, due to the scarcity of data on resilience, burnout, and well-being in the population of Polish medical students, our data only reflect the current situation, and we could not assess if there was any deterioration of the studied parameters. Nevertheless, some similar research projects were performed in Europe; a study from Cyprus, which also used the MBI-SS questionnaire, revealed that burnout prevalence did not differ significantly before and after the pandemic in the population as a whole.

**Table 5. Correlations between Medical Students Well-Being Index (MSWBI) and burnout and resilience scores.**

| MSWBI correlations | Correlation coefficient (r) | P value |
|---|---|---|
| **Burnout** | | |
| Mean emotional exhaustion score | 0.59 | <0.01 |
| Mean cynicism score | 0.48 | <0.01 |
| Mean academic efficacy score | -0.30 | <0.001 |
| **Resilience** | | |
| RS-14 score | -0.35 | <0.01 |

However, it did increase in the group of final-year medical students. In the group of fourth-year medical students, burnout prevalence decreased, we also reported lowest burnout among fourth-year medical students in our study. Emotional exhaustion also decreased in this group, while cynicism increased in all groups [32]. Research on burnout and study satisfaction conducted in Croatia did not reveal the influence of the first lockdown and the switch to online learning on burnout in medical students, or their perception of study satisfaction [33]. Interestingly, a similar study from Kazakhstan showed that burnout syndrome, depression, anxiety, and somatic symptoms decreased after switching to online learning. However, the prevalence of colleague-related burnout during online learning increased in that time [34]. Based on reports from other European countries that examined the level of mental health, cynicism, and burnout in medical students before and during the pandemic, it was concluded that digital learning carries significant risks. It has been found that not only does mental health deteriorate but the level of cynicism also increases. Emotional exhaustion has been found to increase especially in final year students who struggle with a lack of clinical experience just prior to joining qualified junior physicians [32]. In our study, the median MSWBI value was 5 points out of a maximum 7 (greatest distress). This entails less than average performance in psychological and physical domains of life mixed with a lack of awareness of students' own abilities, fatigue, and an inability to cope with stress. It has been reported that anxiety levels among medical students are substantially higher than in the general population [33]. Other research showed that medical students have a high prevalence of distress and depression, especially among 1st-year students, due to issues of adjusting to a new environment and increased workload [34]. In our study, we report the lowest resilience levels in 1st- and 6th-year medical students. It may be hypothesized, therefore, that these subpopulations may suffer most from new environment adjustments (1st-year students) and increased workload combined with new challenges (6th-year students). It was reported that depression and depressive symptoms are common among physicians [35], and their suicide rates are higher than in the general population [36]. So, future doctors with poor well-being joining the workforce during and after the pandemic may have an elevated risk of mental health problems as well. It was confirmed that depression among medical students in Poland is common [37], and due to the lack of central regulations, psychological support is provided by medical schools individually. Those solutions focus on short-term, temporary therapy [38]. The majority of respondents (80.2%) reported feeling that social aversion and mistrust towards doctors increased during the pandemic, and many of them (43.3%) claimed that it may affect their enthusiasm about their future job. Although there is very little research about mistrust among patients in Poland fueled by the COVID-19 pandemic, scientists from other countries reported that many people endorse conspiracy theories about the origins of the pandemic [39], and some are hesitant to be vaccinated because of their distrust of political and medical institutions, "anti-establishment" sentiments, and conspiratorial and paranoid beliefs [40, 41] and, because of the shortage of healthcare professionals [42–44], the need to care for COVID-19 patients and prevent further spread of the virus, the "non-COVID" patients are often left waiting in long queues for telehealth that may not be enough, especially for those with lower socioeconomic backgrounds, who can't afford easy access to the devices needed for telemedicine [45]. All these problems have also been present in Poland and may have contributed to the growth of medical mistrust among polish patients.

The conclusions from our study should encourage both faculty and student organizations to develop proper resilience-building strategies. Besides, there is a strong need to support students' mental health and to monitor students' well-being during the time of recovery from the pandemic. Today's students are the physicians of tomorrow and their ability to adapt to challenges is crucial for effective and efficient patient care in the future.

## Supporting information

**S1 File. Self-created part of the study survey.**
(DOCX)

**S2 File. The study database.**
(XLSX)

## Acknowledgments

The authors are grateful to all medical students who participated in the survey and sincerely thank everyone who shared the survey online.

## Author Contributions

**Conceptualization:** Joanna Forycka, Ewa Pawłowicz-Szlarska, Michał Nowicki.

**Data curation:** Joanna Forycka, Ewa Pawłowicz-Szlarska.

**Formal analysis:** Joanna Forycka, Ewa Pawłowicz-Szlarska, Anna Burczyńska, Natalia Cegielska, Karolina Harendarz.

**Investigation:** Anna Burczyńska.

**Methodology:** Joanna Forycka, Ewa Pawłowicz-Szlarska, Karolina Harendarz.

**Project administration:** Joanna Forycka, Ewa Pawłowicz-Szlarska, Natalia Cegielska, Karolina Harendarz.

**Software:** Joanna Forycka, Anna Burczyńska, Natalia Cegielska.

**Supervision:** Michał Nowicki.

**Validation:** Karolina Harendarz.

**Visualization:** Joanna Forycka, Ewa Pawłowicz-Szlarska, Anna Burczyńska, Natalia Cegielska.

**Writing – original draft:** Joanna Forycka, Ewa Pawłowicz-Szlarska, Anna Burczyńska, Natalia Cegielska.

**Writing – review & editing:** Michał Nowicki.

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
