## [Decision Letter · Decision Letter 0]

6 Oct 2021

PONE-D-21-26801Medical students facing the pandemic – assessment of resilience, well-being and burnout in the COVID-19 eraPLOS ONE

Dear Dr. Nowicki,

Thank you for submitting your manuscript to PLOS ONE. After careful consideration, we feel that it has merit but does not fully meet PLOS ONE’s publication criteria as it currently stands. Therefore, we invite you to submit a revised version of the manuscript that addresses the points raised during the review process.

We look forward to receiving your revised manuscript.

Kind regards,

Stephen Chun

Academic Editor

PLOS ONE

Journal Requirements:

a) Did participants provide their written or verbal informed consent to participate in this study?

“This study was supported by Medical University of Lodz (503/1-151-02/503-01). No external funding was received”

We note that you have provided additional information within the Funding Section that is not currently declared in your Funding Statement. Please note that funding information should not appear in the Acknowledgments section or other areas of your manuscript. We will only publish funding information present in the Funding Statement section of the online submission form.

Reviewers' comments:

Reviewer's Responses to Questions

**Comments to the Author**

1. Is the manuscript technically sound, and do the data support the conclusions?

Reviewer #1: Partly

Reviewer #2: Partly

2. Has the statistical analysis been performed appropriately and rigorously? 

Reviewer #1: I Don't Know

Reviewer #2: Yes

3. Have the authors made all data underlying the findings in their manuscript fully available?

Reviewer #1: No

Reviewer #2: Yes

4. Is the manuscript presented in an intelligible fashion and written in standard English?

Reviewer #1: Yes

Reviewer #2: No

5. Review Comments to the Author

Reviewer #1: The manuscript titled “Medical students facing the pandemic – assessment of resilience, well-being and burnout in the COVID-19 era” assessed resilience, well-being and burnout using validated tools as well as self-created questionnaire administered through Facebook and other online students’ platforms over a one-month period. The majority of medical students had lower levels of resilience and higher burnout with corresponding well-being. Lower resilience and higher burnout was associated with diagnosed mental health conditions, increased use of stimulants, negative self-esteem, and reduced motivation to learn, while higher resilience and less burnout was associated with willingness to work voluntarily in the COVID-19 units.

This is an interesting manuscript that may help fill the gap in information regarding resilience, burnout and well-being in Polish medical students. As the authors suggest, their data only reflects the situation with COVID-19 at the time of collection, and future studies should monitor the situation as the country recovers or changes in the future. It also may provide points for institutions outside of Poland to consider. Listed below are suggestions and comments to help clarify some points and to potentially improve the utility of this manuscript:

General:

• Strongly suggest editing for English grammar and spelling.

Title:

• Line 1: Suggest adding “Polish” to the beginning of the title so it reads: “Polish medical students facing the pandemic – assessment of resilience, well-being and burnout in the COVID-19 era” to be an accurate reflection of the manuscript

Abstract:

• Line 24: Introduction: Suggest replacing “MSs” with “Polish medical students”

• Line 27: Methods: the self-made survey is not mentioned

• Line 34: Results: add “friends and” in front of “relatives”

• General: Mental health was a large component of this manuscript, but it is not mentioned in the abstract. Neither is use of stimulants or views on mistrust towards healthcare professionals.

Introduction:

• There should be reference after every statement – for example, at the end of line 52, it mentions many past studies, but there is no reference(s) at the end of that statement.

• Lines 67-69 is a statement that says “might have taken a toll” – the authors may be able to find a reference that can change the statement to “have taken a toll”. Or if unable to find one, then can state that and bring up the study.

• Would be helpful to understand what the restrictions in Poland were for COVID-19.

• There is a statement defining resilience (lines 95-97) in the methods that may be better placed in the last paragraph of the introduction.

• The last sentence of the introduction doesn’t mention the other factors in the self-created questionnaire about mental health, use of stimulants, presence of COVID-19 infections in respondents and their friends/family, COVID-19 related volunteer or paid work, medical education and views on mistrust towards healthcare professionals.

Methods:

• Line 82: How were the various Facebook groups and studygrams chosen? Was it general medical student group or certain clubs that may introduce a bias in the respondents? Was it possible to determine the response rate, validate that it was medical students who responded, and eliminate duplicates in response (same student answering more than once)?

• Line 83: Were there reminders/multiple posts within the one-month period? How many?

• Line 94: Are the RS-14 levels as seen in Figure 1 defined by the use of the survey? Could you add an label for the x-axis and the score range for each category in the methods section or in the figure?

• Line 101: What are the scores that would indicate poor well-being vs good well-being?

• Line 105: Should specify that the Maslach Burnout Inventory – General Survey for Students was designed for college and university students. Also, the reference is from the 1980s and this particular survey was designed later.

• Line 111: Should add see Table 2 for reference values of low, moderate and high

• Line 112: There should be a statement if two-dimensional burnout for this survey is defined by the survey or by the authors.

• Line 117: How was the self-created part of the survey designed? Did it go through a pilot with students and faculty and refined? Was it a pull down menu of choices or open answer? It would be extremely helpful for the self-created questionnaire to be an appendix of this manuscript. And the data should be provided in accordance to PLOS policy.

• Line 124: It would be helpful if volunteer or paid work was better defined or explained, and why they were combined as they would have different motivating factors. Is volunteer or paid work part of the curriculum?

• Line 141: Please add number of Polish medical schools and whether they all have similar Year 1- Year 6 formats of curriculum. It would also be helpful to have a brief description of the curriculum for those readers who are not familiar with the curriculum.

• Line 144: Is there an estimate of the total number of Polish medical students (and perhaps gender/age/year of study breakdown) to determine an estimated response rate and whether the respondents and their demographics are a representative sample?

Results:

• Lines 162-163: What is the overlap of those two groups? Is there an article about the pre-pandemic levels of mental health issues in this age group?

• Line 168: Is there information regarding how many students were newly diagnosed with psychiatric conditions?

• Line 175-176: Is there information regarding how many students started using stimulants?

• Line 182: In the abstract, it states that “The SARS-CoV-2 infection both among respondents and their relatives did not influence the results”, but this is not stated anywhere else in the manuscript. It could go here. Also, the abstract and methods only mention respondents and family, but Line 180 mentions friends – they should all be consistent. It may also be helpful to understand the context – what was the infection rate and death rate due to COVID-19 in Poland?

• Line 184: Would it be possible to separate out the participants who worked, those who currently work and those who plan to work and those participants who volunteered, those who currently volunteer and those to plan to volunteer? These categories may be different.

• Line 186: What are examples for potential consequences?

• Lines 189-194: Were these choices in the survey or an open answer? It looks like a list that students chose from – what if students had a potential concern that was not listed? The choices and number of respondents could be more clear in table form.

• Line 209: What is the Final Medical Exam? Is this for graduation? Ability to practice? Both?

• Line 220: It is interesting that 80.2% of respondents through that social aversion and mistrust towards doctors increased during the pandemic – this should be discussed in the next section with possible reasons and an explanation of public perceptions in Poland. Can also give examples of challenges faced by the national healthcare system mentioned in line 222.

• Line 266: Replace “particular” with “the three”

Discussion:

• Line 324: Why do the authors think that the lowest resilience in among 1st and 6th year students? Are there articles to support similar findings or are these unique?

• Line 344: Although there is scarce data on resilience, burnout and well-being in the population of Polish medical students, are there reports from nearby countries or those with similar curriculum that were similarly affected by COVID? How are the results similar or unique?

• As mentioned above for Line 220: It is interesting that 80.2% of respondents through that social aversion and mistrust towards doctors increased during the pandemic – this should be discussed in the next section with possible reasons and an explanation of public perceptions in Poland.

• Could add a statement that the study population is outside the validated range of RS-14.

• There is a lot of data presented, but the conclusion is to develop resilience building strategies. What about mental health support?

Tables and Figures:

• Suggest a figure showing average resilience score in Years 1 through 6.

• Suggest a figure showing average scores for the three dimensions of burnout and frequency of two- and three-dimensional burnout in Years 1 through 6.

• Suggest add graph of RS-14 score and two-dimensional burnout and RS-14 score and well-being score in Figure 2.

• Suggest a distribution figure showing well-being scores, similar to resilience, and average well-being scores in Years 1 through 6.

• Could Table 3 be a scatter plot, or a more visual way to show the distribution? Is there a correlation coefficient between burnout and resilience? How similar are these?

Reviewer #2: 1. Standard English should be used for the entire manuscript. Specific sentences that were unclear include Lines 44, 52, 58, 63, 68-69, 111.

2. Lines 52-55 refer to “many past studies”, however there are no citations associated with this statement.

3. Lines 59-60 of the introduction states findings “reported in our study”, however this would be more appropriate in the discussion. Additionally, the sentence regarding “decreased motivation and concerns of inappropriate levels of knowledge, etc.” should be supported by a reference.

4. Resilience is defined in two areas- lines 70 and again in line 95-96, which is redundant.

5. In the Methods section, Lines 117-135 when explaining the self-created part of the survey, it may be better to summarize the five question domains in the text, then provide the actual questions on the survey in supplement material.

6. Line 142- Is it supposed to be “medical facilities” instead of “medical faculties”?

7. Lines 144-146 and Table 1 present survey results and should be moved to the Results section.

8. Lines 157 and 286- chi-square should be written out explicitly

9. What is the response rate to the survey? The data presented in Line 144 describes that 1864 respondents entered the survey, however, how many total medical students are there in Poland who could have completed the survey?

10. Results section- Lines 162-225 describe all of the results of the self-created survey. These results may be better presented in a table and the key findings highlighted in the text of the results section.

11. Figure 1 should have a label on the X-axis

12. It is unclear how the “resilience score” correlates with the 6 categories of resilience in Figure 1 (very low, low, on the low end, moderate, moderately high, high). Placing a legend in the figure may help clarify this.

13. Results section- Lines 236-262 describe the correlation between the resilience score and medical student characteristics gathered in the self-created survey. These results may be better presented in a table and the highlights discussed in the results section.

14. Results section- Lines 280-287 describe the correlation between burnout and medical student characteristics gathered in the self-created survey. These results may be better presented in a table and the highlights discussed in the results section.

15. Figure 2- the meaning of the 2 or 3 stars/dots in the figure is unclear. Please explain

16. Line 296-297- The description of the MSWBI score range and meaning should be in the methods section.

17. Lines 319-321- This was a one-time survey completed during the pandemic, therefore, you don’t have a baseline measure of resilience, burn out and well-being before the pandemic. Therefore, the statement that “2/3 of the respondents presented lower levels of resilience indicated a problem with ability to maintain or regain mental health while experience adversity such as a global pandemic” is not possible because you don’t know what their baseline level of resilience was before the pandemic.

18. Lines 324-326- What are potential reasons the 1st and 6th year students had the lowest resilience?

19. The Discussion is limited to a discussion about resilience and suggestions to improve education about resilience in medical education. Please expand the discussion to also comment on burnout and well-being.

20. Please expand on other limitations of the study aside from the survey being online and the author bias of the self-created survey questions.

6. PLOS authors have the option to publish the peer review history of their article (what does this mean?). If published, this will include your full peer review and any attached files.

Reviewer #1: No

Reviewer #2: No

---

## [Author Response · Author response to Decision Letter 0]

4 Dec 2021

Responses to Reviewer’s 1 comments

General 

#1 Comment:

Strongly suggest editing for English grammar and spelling.

#1 Response:

Thank you for this remark. The manuscript was carefully edited for grammar and spelling by English native speaker.

Title 

#1 Comment:

Line 1: Suggest adding “Polish” to the beginning of the title so it reads: “Polish medical students facing the pandemic – assessment of resilience, well-being and burnout in the COVID-19 era” to be an accurate reflection of the manuscript

#1 Response:

Thank you for this remark. The suggested change of the title has been applied for more accurate reflection of the scope of the manuscript. 

Abstract 

#1 Comment:

Line 24: Introduction: Suggest replacing “MSs” with “Polish medical students”

#1 Response:

Thank you for this remark. The suggested change has been applied in Introduction. 

#2 Comment:

Line 27: Methods: the self-made survey is not mentioned. 

#2 Response:

Thank you for this comment. The Information about self-created part of the survey has been added in the abstract. 

#3 Comment:

Line 34: Results: add “friends and” in front of “relatives”

#3 Response:

Thank you for this correction, the change has been applied as suggested.

#4 Comment:

General: Mental health was a large component of this manuscript, but it is not mentioned in the abstract. Neither is use of stimulants or views on mistrust towards healthcare professionals.

#4 Response:

Thank you for this remark, information on the above mentioned issues has been added in the Results section of the abstract. 

Introduction

#1 Comment:

There should be reference after every statement – for example, at the end of line 52, it mentions many past studies, but there is no reference(s) at the end of that statement.

#1 Response:

Thank you for this valuable comment, the appropriate references have been added. 

#2 Comment:

Lines 67-69 is a statement that says “might have taken a toll” – the authors may be able to find a reference that can change the statement to “have taken a toll”. Or if unable to find one, then can state that and bring up the study.

#2 Response:

Unfortunately, we were not able to find such reference. We have changed the sentence, so now it reads: “It may be hypothesized, that these extraordinary actions such as social distancing and closing schools, offices and entertainment venues might have taken a toll especially on more susceptible to stress and less resilient individuals.” Hopefully, it can be accepted in that form. 

#3 Comment:

Would be helpful to understand what the restrictions in Poland were for COVID-19.

#3 Response:

Thank you for this suggestion. The information about COVID-19-related restrictions in Poland has been added. 

#4 Comment:

There is a statement defining resilience (lines 95-97) in the methods that may be better placed in the last paragraph of the introduction.

#4 Response:

Thank you very much for this remark. Indeed, definitions of all studied parameters fit better in the Introduction section. The correction has been applied – the definition is deleted from the Methods. 

#5 Comment:

The last sentence of the introduction doesn’t mention the other factors in the self-created questionnaire about mental health, use of stimulants, presence of COVID-19 infections in respondents and their friends/family, COVID-19 related volunteer or paid work, medical education and views on mistrust towards healthcare professionals.

#5 Response:

Thank you for this suggestion. The lacking information has been added. 

Methods

#1 Comment:

Line 82: How were the various Facebook groups and studygrams chosen? Was it general medical student group or certain clubs that may introduce a bias in the respondents? Was it possible to determine the response rate, validate that it was medical students who responded, and eliminate duplicates in response (same student answering more than once)?

#1 Response: 

Thank you for these questions. The survey was posted on national or local Facebook groups for medical students, all of them were open for all polish medical students or students from the particular institution. Also, we contacted student influencers running the nationwide Instagram profiles focused on studying medicine and asked them to share our survey. Via this channel survey was available for all subscribers of the particular profile. Unfortunately, due to character of the survey (online survey) it was not possible to estimate the response rate, since we did not monitor number of subscribers of particular profiles at the time when the survey was open. Also, during this month, these numbers could change. Due to the character of data collection (100% anonymous, no computer IP number collection) we could not prevent the same person filling out the questionnaire twice. Answers to these questions are now added in the Methods section. 

#2 Comment:

Line 83: Were there reminders/multiple posts within the one-month period? How many?

#2 Response: 

No, the survey was posted once at all of the above-mentioned groups. This information is now added in the Methods section.

#3 Comment:

Line 94: Are the RS-14 levels as seen in Figure 1 defined by the use of the survey? Could you add an label for the x-axis and the score range for each category in the methods section or in the figure?

#3 Response: 

Thank you for this suggestion. The x-axis label was added in Figure 1. Yes, resilience levels are defined by the survey authors and we used these levels in the study. The score range for each category has been added in the Methods section. 

#4 Comment:

Line 101: What are the scores that would indicate poor well-being vs good well-being?

#4 Response: 

Thank you for this question. There is no such clear specification of poor and good well-being according to this survey. It said in the survey manual that MSWBI score ranges from 0 to 7, and 7 points indicate the greatest distress. This clarification is now added in the Methods section and deleted from the Results section. 

#5 Comment:

Line 105: Should specify that the Maslach Burnout Inventory – General Survey for Students was designed for college and university students. Also, the reference is from the 1980s and this particular survey was designed later.

#5 Response: 

Thank you for this remark. The specification has been added. Indeed, the reference referred to burnout itself, and the proper citation should refer to the survey, which was proposed in the 4th version of the manual by Maslach et al. in 2017 (Maslach, C., Jackson, S. E., and Leiter, M. P. (2017). Maslach Burnout Inventory Manual, 4th Edn. Menlo Park, CA: Mind Garden). The citation is now added accordingly. 

#6 Comment:

Line 111: Should add see Table 2 for reference values of low, moderate and high

#6 Response: 

Thank you for this remark. The correction has been applied in line with the suggestion. 

#7 Comment:

Line 112: There should be a statement if two-dimensional burnout for this survey is defined by the survey or by the authors.

#7 Response: 

The two-dimensional burnout (high cynicism + high emotional exhaustion) has been already described in previous studies in the field. However, three-dimensional criterion is most frequently adopted in studies. It is now explained in the Methods section. 

#8 Comment:

Line 117: How was the self-created part of the survey designed? Did it go through a pilot with students and faculty and refined? Was it a pull down menu of choices or open answer? It would be extremely helpful for the self-created questionnaire to be an appendix of this manuscript. And the data should be provided in accordance to PLOS policy.

#8 Response:

Thank you for this remark. As it is suggested we have added the self-created part of the survey as the supplementary material. Yes, the pilot was performed in the group of 25 students of different years and their remarks on the questions quality were applied in the final version of the survey, this information is now added in the Methods section. 

#9 Comment:

Line 124: It would be helpful if volunteer or paid work was better defined or explained, and why they were combined as they would have different motivating factors. Is volunteer or paid work part of the curriculum?

#9 Response: 

Thank you for this remark. The extracurricular volunteer and paid work at the time of COVID-19 pandemic is now explained in more detail in the Methods section. 

#10 Comment:

Line 141: Please add number of Polish medical schools and whether they all have similar Year 1- Year 6 formats of curriculum. It would also be helpful to have a brief description of the curriculum for those readers who are not familiar with the curriculum.

#10 Response: 

Thank you for this suggestion. The number of Polish medical schools has been added. Also, a brief description of the curriculum, which is the same at all Polish medical schools, is provided. 

#11 Comment:

Line 144: Is there an estimate of the total number of Polish medical students (and perhaps gender/age/year of study breakdown) to determine an estimated response rate and whether the respondents and their demographics are a representative sample?

#11 Response: 

Thank you for this question. The estimate of the total number of Polish medical students of all years in academic year 2019/2020 was around 32 thousands. Unfortunately, there are no open access data on demographics of this group and determination whether the respondents are a representative sample is not possible. 

Results

#1 Comment:

Lines 162-163: What is the overlap of those two groups? Is there an article about the pre-pandemic levels of mental health issues in this age group?

#1 Response:

Thank you for this valuable question. There were 203 students who did not seek psychological help in the past, but felt such need at the time of pandemic, this information is now added in the Results section. Indeed, we found two paper on mental health issues in the population of polish medical students in the pre-pandemic era, both are now cited in the Discussion section. 

#2 Comment:

Line 168: Is there information regarding how many students were newly diagnosed with psychiatric conditions?

#2 Response: 

Thank you for this question. No, our survey did not include the question of how many students were newly diagnosed with psychiatric conditions.

#3 Comment:

Line 175-176: Is there information regarding how many students started using stimulants?

#3 Response:

Thank you for this question. No, we did not ask in the survey about start of using stimulants during the pandemic. 

#4 Comment:

Line 182: In the abstract, it states that “The SARS-CoV-2 infection both among respondents and their relatives did not influence the results”, but this is not stated anywhere else in the manuscript. It could go here. Also, the abstract and methods only mention respondents and family, but Line 180 mentions friends – they should all be consistent. It may also be helpful to understand the context – what was the infection rate and death rate due to COVID-19 in Poland?

#4 Response: 

Thank you for this valuable comment. Now all the parts are consistent with regard to this issue. Also, the information about infection and death rates due to COVID-19 in Poland has been added. 

#5 Comment:

Line 184: Would it be possible to separate out the participants who worked, those who currently work and those who plan to work and those participants who volunteered, those who currently volunteer and those to plan to volunteer? These categories may be different.

#5 Response:

Thank you for this remark, the correction has been applied.

#6 Comment:

Line 186: What are examples for potential consequences?

#6 Response:

The potential consequences are now added in Results section. 

#7 Comment:

Lines 189-194: Were these choices in the survey or an open answer? It looks like a list that students chose from – what if students had a potential concern that was not listed? The choices and number of respondents could be more clear in table form.

#7 Response: 

Thank you for this remark. There were choices in the survey and the list of potential answers was based on the discussions with the participants of the pilot period of the study. Due to huge estimated number of participants we decided not to introduce open answers in this questions, because of potential problems with further analysis. In line with the Reviewer’s suggestion it is now presented in the Table form (Table 2). 

#8 Comment:

Line 209: What is the Final Medical Exam? Is this for graduation? Ability to practice? Both?

#8 Response: 

Thank you for these questions. The explanation about the Exam is now provided. 

#9 Comment: 

Line 220: It is interesting that 80.2% of respondents through that social aversion and mistrust towards doctors increased during the pandemic – this should be discussed in the next section with possible reasons and an explanation of public perceptions in Poland. Can also give examples of challenges faced by the national healthcare system mentioned in line 222.

#9 Response: 

Thank you for this comment. Both possible explanation of the high rate of mistrust towards doctors and examples of challenges faced by Polish healthcare system are now provided accordingly. 

#10 Comment:

Line 266: Replace “particular” with “the three”

#10 Response:

Thank you for this correction. It has been applied in the manuscript. 

Discussion

#1 Comment: 

Line 324: Why do the authors think that the lowest resilience in among 1st and 6th year students? Are there articles to support similar findings or are these unique?

#1 Response: 

Thank you for this remark. The lowest resilience levels were found in 1st and 6th year students of our study group and we described this in the Results section – “Resilience was not dependent on age; however, the highest resilience level was found among 4th year students (68.3±12.9 points), and the lowest among 1st (62.9±12.9) and 6th year students (62.6±14.4); p<0.01.” Now, it is replaced with the Figure 2. in line with the Reviewer’s suggestion. We did not find any other studies particularly addressing resilience in Years 1 through 6. However, there is a study indicating that medical students have a high prevalence of distress and depression, especially the 1st year students due to issues of adjustment to new environment and increased workload. This citation has been added in the Discussion section. 

#2 Comment: 

Line 344: Although there is scarce data on resilience, burnout and well-being in the population of Polish medical students, are there reports from nearby countries or those with similar curriculum that were similarly affected by COVID? How are the results similar or unique?

#2 Response: 

Thank you for this suggestion. The studies from Cyprus, Croatia and Kazakhstan are now cited and discussed in the Discussion section. 

#3 Comment: 

As mentioned above for Line 220: It is interesting that 80.2% of respondents through that social aversion and mistrust towards doctors increased during the pandemic – this should be discussed in the next section with possible reasons and an explanation of public perceptions in Poland.

#3 Response: 

Thank you for that remark. As mentioned above, the suggested explanation is now provided in the Discussion section. 

#4 Comment:

Could add a statement that the study population is outside the validated range of RS-14.

#4 Response: 

Thank you for this remark. As it is mentioned in the Methods section, The RS-14 is validated in the population of young Polish adults (aged 19-27). In our opinion, the population of Polish medical students can be treated as a part of the “population of young Polish adults”. In our study group there were only 52 respondents (2.8%) who reported there age as “older than 26 years”. 

#5 Comment:

There is a lot of data presented, but the conclusion is to develop resilience building strategies. What about mental health support?

#5 Response: 

Thank your very much for this valuable remark. We have corrected the conclusion in line with the suggestion. 

Tables and Figures

#1 Comment:

Suggest a figure showing average resilience score in Years 1 through 6.

#1 Response: 

Thank you for this suggestion. We have prepared the figure showing average resilience score in Years 1 through 6 (Figure 2). 

#2 Comment: 

Suggest a figure showing average scores for the three dimensions of burnout and frequency of two- and three-dimensional burnout in Years 1 through 6.

#2 Response: 

Thank you for this suggestion. The figure has been added. 

#3 Comment: 

Suggest add graph of RS-14 score and two-dimensional burnout and RS-14 score and well-being score in Figure 2.

#3 Response: 

Thank you for the suggestion. Figure 2. was meant to show relation between burnout and resilience. We did not intend to show two-dimensional burnout there as parameter less frequently used in the literature. Relation between resilience and well-being is presented in the text in the Results section. 

#4 Comment: 

Suggest a distribution figure showing well-being scores, similar to resilience, and average well-being scores in Years 1 through 6.

#4 Response: 

Thank you for this suggestion. Median MSWBI score for years 1-3 and 6 is 5, and for years 4 and 5 – 4, no significant differences were found with regard to it. This information has been added in the Results section. 

#5 Comment: 

Could Table 3 be a scatter plot, or a more visual way to show the distribution? Is there a correlation coefficient between burnout and resilience? How similar are these?

#5 Response: 

Thank you for this remark. Indeed, there is a significant negative correlation between burnout and resilience (r=-0.38, p<0.05), this information is added in the Results section. The scatter plot for MSWBI which takes integer values from 0 to 7 would not be very informative, that is way we decided to show these data in the table. 

Responses to Reviewer’s 2 comments

#1 Comment: 

Standard English should be used for the entire manuscript. Specific sentences that were unclear include Lines 44, 52, 58, 63, 68-69, 111.

#1 Response: 

Thank you for this remark. The manuscript was edited for grammar and spelling by English native speaker. 

#2 Comment: 

Lines 52-55 refer to “many past studies”, however there are no citations associated with this statement.

#2 Response: 

Thank you very much for this remark. The reference to the review article has been added in line with the suggestion. 

#3 Comment: 

Lines 59-60 of the introduction states findings “reported in our study”, however this would be more appropriate in the discussion. Additionally, the sentence regarding “decreased motivation and concerns of inappropriate levels of knowledge, etc.” should be supported by a reference.

#3 Response: 

Thank you for this remark. The sentence has been deleted from the Introduction. 

#4 Comment: 

Resilience is defined in two areas- lines 70 and again in line 95-96, which is redundant.

#4 Response: 

Thank you very much for this suggestion. The definition of resilience was deleted from the Methods section. 

#5 Comment: 

In the Methods section, Lines 117-135 when explaining the self-created part of the survey, it may be better to summarize the five question domains in the text, then provide the actual questions on the survey in supplement material.

#5 Response: 

Thank you for this suggestion. For the better understanding of all parameters that we studied we would like to leave the description of this part of the survey in the text but additionally we have prepared the self-created part of the survey as the supplemental material. 

#6 Comment: 

Line 142- Is it supposed to be “medical facilities” instead of “medical faculties”?

#6 Response:

Thank you for this remark. The term “medical faculties” is a term used to describe the faculty of medicine, which is a part of a “non-medical” University. Medicine is taught as an undergraduate, 6-year degree in Poland and it is possible to study medicine at the medical university (such as Medical University of Lodz) which offers only medicine and other medical sciences degrees. The other option is to study at the medical faculty of an university that also offers a variety of courses besides medical ones such as law, economy, engineering, management etc. Such examples are the Faculty of Medicine of the Jagiellonian University in Krakow or the Faculty of Medical Sciences and Health Sciences of the Kazimierz Pulawski University of Technology and Humanities in Radom. This is why we used the term “medical faculties” and we believe no changes or correction are necessary with that regard. 

#7 Comment: 

Lines 144-146 and Table 1 present survey results and should be moved to the Results section.

#7 Response:

Thank you for this remark. Since both information on the percent of participant who answered all questions and study group characteristics (Table 1.) describe the study group we would like to ask to consider leaving these in the Methods and participants section of the manuscript. 

#8 Comment: 

Lines 157 and 286- chi-square should be written out explicitly

#8 Response:

Thank you, the correction has been applied. 

#9 Comment: 

What is the response rate to the survey? The data presented in Line 144 describes that 1864 respondents entered the survey, however, how many total medical students are there in Poland who could have completed the survey?

#9 Response:

Thank you for this question. The estimate of the total number of Polish medical students of all years in academic year 2019/2020 was around 32 thousands. Unfortunately, due to character of the survey (online survey) it was not possible to estimate the response rate, since we did not monitor number of subscribers of particular profiles (Instagram) and members of particular Facebook groups at the time when the survey was open. Also, during this month, these numbers could change. 

#10 Comment: 

Results section- Lines 162-225 describe all of the results of the self-created survey. These results may be better presented in a table and the key findings highlighted in the text of the results section.

#10 Response:

Thank you for this comment. Indeed, there is much data in the text. We tried to present it in the table but there are totally different issues raised in particular part of the text referring to the factors that we assessed in the survey, so one table would not be possible. This could lead to plenty small tables. At the same time, we believe that presenting all this data gives better background for the reader and better describes the studied population. We have added the table on potential concerns of students regarding the voluntary work in the COVID-19 health-care units. 

#11 Comment: 

Figure 1 should have a label on the X-axis

#11 Response:

Thank you for this remark. The x-axis label has been added in Figure 1. 

#12 Comment: 

It is unclear how the “resilience score” correlates with the 6 categories of resilience in Figure 1 (very low, low, on the low end, moderate, moderately high, high). Placing a legend in the figure may help clarify this.

#12 Response:

Thank you for this suggestion. The score ranges for each resilience level have been added in the Methods section.

#13 Comment: 

Results section- Lines 236-262 describe the correlation between the resilience score and medical student characteristics gathered in the self-created survey. These results may be better presented in a table and the highlights discussed in the results section.

#13 Response:

Thank you very much for this remark. Some of these results are presented in the figure (Figure 2), the rest is presented in the table (Table 3). 

#14 Comment: 

Results section- Lines 280-287 describe the correlation between burnout and medical student characteristics gathered in the self-created survey. These results may be better presented in a table and the highlights discussed in the results section.

#14 Response:

Thank you for this remark. Due to different types of variables we had to use different tests to compare these parameters (Chi-square and t-test). This could lead to confusion if we present these two types of variables in one table. 

#15 Comment: 

Figure 2- the meaning of the 2 or 3 stars/dots in the figure is unclear. Please explain

#15 Response:

Thank you for this remark. The stars refer to the p values. The clarification has been added in the legend of the figure. 

#16 Comment: 

Line 296-297- The description of the MSWBI score range and meaning should be in the methods section.

#16 Response:

Thank you for this remark. The correction has been applied. 

#17 Comment: 

Lines 319-321- This was a one-time survey completed during the pandemic, therefore, you don’t have a baseline measure of resilience, burn out and well-being before the pandemic. Therefore, the statement that “2/3 of the respondents presented lower levels of resilience indicated a problem with ability to maintain or regain mental health while experience adversity such as a global pandemic” is not possible because you don’t know what their baseline level of resilience was before the pandemic.

#17 Response:

Thank you for this remark. By using “lower levels” we did not mean “lower than before the pandemic” but “on the lower end of the scale” (‘very low’, ‘low’, ‘on the lower end’) what may be confusing. We have corrected the sentence so now it reads: “2/3 of the respondents presented low levels of resilience indicating a problem with ability to maintain or regain mental health while experience adversity such as a global pandemic”. 

#18 Comment: 

Lines 324-326- What are potential reasons the 1st and 6th year students had the lowest resilience?

#18 Response:

Thank you for this question. Potential reasons the 1st and 6th year students had the lowest resilience are now provided in the Discussion section. 

#19 Comment: 

The Discussion is limited to a discussion about resilience and suggestions to improve education about resilience in medical education. Please expand the discussion to also comment on burnout and well-being.

#19 Response:

Thank you for this suggestion. The discussion has been expanded as suggested. 

#20 Comment: 

Please expand on other limitations of the study aside from the survey being online and the author bias of the self-created survey questions.

#20 Response:

Thank you for the suggestion. The most important limitation is no possibility to compare our results with the pre-pandemic landscape, what is enumerated in the Discussion section. Besides, sampling and desirability biases are explained in more detail with regard to our study.

---

## [Editor Report · Decision Letter 1]

9 Dec 2021

Polish medical students facing the pandemic – assessment of resilience, well-being and burnout in the COVID-19 era

PONE-D-21-26801R1

Dear Dr. Nowicki,

We’re pleased to inform you that your manuscript has been judged scientifically suitable for publication and will be formally accepted for publication once it meets all outstanding technical requirements.

Kind regards,

Stephen Chun

Academic Editor

PLOS ONE
---

## [Editor Report · Acceptance letter]

14 Jan 2022

PONE-D-21-26801R1 

Polish medical students facing the pandemic – assessment of resilience, well-being and burnout in the COVID-19 era 

Dear Dr. Nowicki:

I'm pleased to inform you that your manuscript has been deemed suitable for publication in PLOS ONE. Congratulations! Your manuscript is now with our production department. 

Kind regards, 

on behalf of

Dr. Stephen Chun 

Academic Editor

PLOS ONE